# Studying the Research–Practice Gap in Physical Therapies for Cerebral Palsy: Preliminary Outcomes Based on a Survey of Spanish Clinicians

**DOI:** 10.3390/ijerph192114535

**Published:** 2022-11-05

**Authors:** Cristina Sanchez, Sergio Lerma-Lara, Rodrigo Garcia-Carmona, Eloy Urendes, Paula Laccourreye, Rafael Raya

**Affiliations:** 1Departamento de Tecnologías de la Información, Escuela Politécnica Superior, Universidad San Pablo-CEU, CEU Universities, 28668 Madrid, Spain; 2Motion in Brains Research Group, Centro Superior de Estudios Universitarios La Salle, Universidad Autónoma de Madrid, 28023 Madrid, Spain

**Keywords:** cerebral palsy, physical therapies, physiotherapy, occupational therapy, postural control, trunk control, motor control, rehabilitation, survey, clinical practice, scientific evidence

## Abstract

The purpose of this work is to study the gap between the research evidence and the clinical practice in the physical rehabilitation of people with cerebral palsy. A review process was performed to (1) identify physical therapies to improve postural control in children with cerebral palsy and (2) determine the scientific evidence supporting the effectiveness of those therapies. A Likert-based survey addressing a total of 43 healthcare professionals involved in pediatric physical therapy departments in Spain was carried out. The discussion was mainly supported by studies of level I or II evidence (according to the Oxford scale). The search process yielded 50 studies reporting 16 therapies. A strong positive correlation between the most used treatments and elevated levels of satisfaction was found. Some well-known but not often used techniques, such as hippotherapy, were identified. The treatment with the highest degree of use and satisfaction—neurodevelopment therapy (Bobath)—and some emerging techniques, such as virtual reality, were also identified. The fact that there is a meaningful gap between clinical practice and the scientific evidence was confirmed. The identified gap brings a certain degree of controversy. While some classic and well-known therapies had poor levels of supporting evidence, other relatively new approaches showed promising results.

## 1. Introduction

Cerebral palsy (CP) is a well-known neurodevelopmental condition beginning in early childhood and persisting throughout the lifespan. It is defined as a “a group of permanent disorders of the development of movement and posture, causing activity limitation, that are attributed to non-progressive disturbances that occurred in the developing fetal or infant brain” [1]. CP has traditionally been classified according to the type of damage (spasticity, hypotonia, dyskinesia or ataxia) and its topographical distribution (unilateral CP or hemiplegia, and bilateral CP described as diplegia or tetraplegia) [2].

According to the World Health Organization’s (WHO) International Classification of Functioning, Disability and Health (ICF)—a framework for measuring health and disability at both the individual and population levels, which was officially endorsed by all 191 Member States in 2001 (resolution WHA 54.21) as the international standard to describe and measure health and disability—clinicians should focus the intervention on the person’s ‘functioning’ and autonomy in daily living.

The Gross Motor Function Classification System (GMFCS) was developed to classify functional mobility in children diagnosed with CP by the level of motor function in lower limbs. It describes five levels ranging from level I, indicating children with minimal or no mobility dysfunction compared to the general population, to level V, covering children who are dependent and need technical aids to move [3]. Regarding the prevalence of CP in developed countries, 2–3 cases per 1000 live births are estimated [4]. Moreover, CP is considered one of the most common causes of physical disability in children [5].

In addition to several well-known consequences of brain damage, including learning disability, speech and language disorders, orthopedic complications, and epilepsy, people with CP usually develop postural and motor abnormalities, which give rise to different degrees of postural control dysfunction [6]. Furthermore, the consequences derived from chronic muscle imbalance results in increasing disability with age.

Postural control can be defined as the ability to control the body’s position in space for purposes of stability and orientation [7,8,9] and depends on the capacities of the neuromuscular and musculoskeletal systems. As their neuromuscular system has restricted capacity for coordinating muscles in postural synergies, this fact gives rise to multiple dysfunctions in sequencing, in activation time for postural response, and in adapting posture to the setting. As a result, these patients have different limitations in gross motor skills requiring balance such as gait, upper limb activities or oral motor activities [8,10].

These limitations clearly influence and restrict a wide range of daily activities such as self-care or education. However, although there is currently a broad spectrum of physical interventions designed to treat this core deficit, and children with CP usually receive or participate in many passive or active interventions aimed to improve movement and posture, often the real impact on postural control is not well-measured or documented, and some of them have even been proven to be ineffectual or unnecessary [8,11,12,13,14,15].

Previous studies [16,17] confirmed that, unfortunately, there is a research–practice gap. In fact, the reality of the clinical practice shows a great disparity of criteria regarding the type of interventions for treating a specific problem in children with CP, depending on the clinical team in charge, which results in a worrying lack of unanimity in the training criteria and important differences in the daily practice of pediatric physiotherapy. In addition, the exposure of both professionals in the field and families of patients with CP to non-evidence-based practices represents another obstacle to consider [18,19]. Within this context, in Spain, the differences in the practice and training of pediatric physiotherapists, together with the need and demand for constant training and updating, motivated the creation of the Sociedad Española de Fisioterapia en Pediatría (SEFIP), whose objectives include the support of evidence-based practice. Therefore, a joint strategy aimed at reducing this gap would lead to more cost-effective interventions, better information given to families and, overall, empower the quality of life of the people with CP.

The vast volume of research published makes it difficult to distinguish the most effective interventions. However, several works have studied the level of evidence of these interventions, and they should be useful tools for clinical practice. Novak et al. [15] conducted a systematic review of 64 interventions divided into three categories: spasticity management, contracture management, and improved muscle strength. According to the authors, only 15 interventions were green light interventions, meaning that the authors recommended using them in clinical practice. Likewise, Dewar et al. [8] performed a systematic review reporting 13 exercise interventions with postural control outcomes for children with CP. Five interventions were supported by a moderate level of evidence, six had weak or conflicting evidence, one was classified as an ineffective intervention and the last one lacked high-level evidence. Both review-based articles used “The Oxford 2011 Levels of Evidence” [20]. In addition, Novak used the GRADE system (endorsed by the World Health Organization) [21] and the Evidence Alert Traffic Light System [22], and Dewar used Oxam and Guyatt’s classification [23], which yields a score out of 10.

All in all, the question (based on the PICo framework for qualitative studies) that motivates this study is: what are the differences between the most-used interventions in pediatric physical therapy in children with CP in Spain and the actual scientific evidence regarding their effectiveness? To answer this question, the purpose of this work (in line with the objectives of the SEFIP) is to study and discuss the gap between the research evidence and clinical practice in the physical rehabilitation of people with CP in Spain, in three main steps. The first step consisted of identifying those physiotherapy techniques, therapies and interventions currently used to improve postural control in children with CP. The second step consisted of analyzing those previously identified methods according to the experience-based opinions and ideas obtained from a survey using the Likert scale addressed to clinical professionals involved in the rehabilitation of people with CP. The survey, performed in Spain, included general information about the participants, questions about the knowledge, the degree of use and the satisfaction of those techniques, therapies and interventions, and an open-ended question to identify alternative treatments and emerging research lines in the rehabilitation of CP. The last step consisted of discussing the differences between the clinical practice preferences and the scientific evidence of effectiveness, defined as the ability of an intervention to provide benefits during usual conditions of clinical care [8], regarding those previously identified methods.

## 2. Materials and Methods

A review of techniques, therapies, and interventions for improving postural control in children with CP was performed. The review process was focused on works published between 2010 and 2021 to have access to the latest advances, considering the work of Novak et al. [15] as the fundamental pillar and main reference. The following databases were searched: PubMed, the Cochrane Library and PEDro. The search of published studies was performed in 2021. Interventions and keywords for investigation were identified using (1) the contributing author’s knowledge of the field; (2) nationally and internationally recognized CP websites such as the Surveillance of Cerebral Palsy in Europe (https://eu-rd-platform.jrc.ec.europa.eu/scpe_en (accessed on 2 November 2022)), the Cerebral Palsy Foundation (https://cpresource.org/ (accessed on 2 November 2022)), the Cerebral Palsy Alliance (www.cerebralpalsy.org.au (accessed on 2 November 2022)), and the Sociedad Española de Fisioterapia en Pediatría (www.sepif.org (accessed on 2 November 2022)); and (3) the top 25 hits in Google using the search term ‘cerebral palsy’, to target the population of interest, and terms such as ‘postural control’, ‘techniques’, ‘treatments’ ‘methods’, ‘interventions’ or ‘rehabilitation’, to target the postural control rehabilitation processes.

In order to identify those physiotherapy techniques, therapies and interventions currently used to improve postural control in children with CP, articles were included if (1) they were full articles published in English, (2) they exclusively included participants aged between 0 to 18 years old (children) with CP regardless of the type and level of dysfunction, (3) the performed therapies and interventions in the target population involved land-based exercises and physiotherapy techniques, and (4) the outcome measures were focusing on assessing postural control of either postural stability or postural orientation. Works were excluded if they were (1) opinion articles, (2) if they reported water-based interventions, or (3) medical or surgical interventions.

The level of evidence was not an exclusion criterion for this first step, including from systematic reviews of randomized trials (level I according to The Oxford 2011 Levels of evidence) to case studies or mechanism-based reasoning (levels IV and V). This decision was taken because we did not want to dismiss methods that could be popular in clinical practice regardless of their levels of evidence.

On the other hand, the discussion is mainly supported by studies of level I or II evidence. Levels III to IV were included only if the highest level of evidence did not exist on the topic.

This project was conducted with the collaboration of the “Hospital Infantil Universitario Niño Jesús”, which is a national reference centre in the rehabilitation of children with cerebral palsy. The research group of this hospital in turn sought collaborators from other centres also specialized in cerebral palsy with which the hospital works regularly, with the aim of sending the survey to physiotherapists/pediatric physiotherapists (1) and occupational/neuro-psychomotricity therapists (2) with fully demonstrated experience (certified by the clinical directors of the institutions involved) in the treatment of children with CP at all levels of the disease involvement, covering the entire spectrum of motor impairment described in the GMFCS. These profiles were included because, even though the first (1) specialize in working with the function and the second (2) in working with the activity and the participation of the users, they both collaborate and coordinate their performance in multidisciplinary clinical teams. A total of 43 people participated in the survey.

The survey (see Appendix A) consisted of five parts:General information about the participants, including age, education or professional career, number of years of experience and working institution.Knowledge of the different treatments. Participants answered if they knew (Yes) or not (No) about each one of the listed techniques, therapies, and interventions.Degree of use of the treatments using a Likert scale. Participants had to choose among six different options depending on how often they had used each one of the listed techniques, therapies, and interventions throughout their careers, according to the following scale: always, very often, sometimes, almost never, never, N/A.Degree of satisfaction of the treatments using a Likert scale. Participants had to choose among six different options based on their degree of satisfaction with each one of the listed techniques, therapies, and interventions after its use on children with cerebral palsy throughout their careers, according to the following scale: very satisfied, satisfied, neither satisfied nor dissatisfied, dissatisfied, very dissatisfied, N/A.The last part of the survey was an open-ended question asking participants to state which treatments, both from the listed ones or from other sources according to their professional experience, they thought had more future in the rehabilitation of this type of patients. Participants had a comment box with no specific pre-set to answer the previous question, and an extra comment box to add extra observations if considered.

This structure, as well as the questions of the survey, were validated by the research group of the “Hospital Infantil Universitario Niño Jesús”. All their recommendations about the clarity of the questions and their relevance were considered and included in the final version of the survey that was subsequently sent to the collaborating institutions.

Once the survey data were recollected, a series of description tables and histograms were made so that the data could be more clearly interpreted. In addition, to compare extensively used treatments and treatments with a high level of satisfaction, both the bivariate correlation (Pearson correlation coefficient) and their corresponding scatter plot and regression line were obtained.

## 3. Results

Using the search strategy, 153 citations were identified, of which 50 met the inclusion criteria, highlighting the systematic reviews of Novak [15] and Dewar [8], where a total of 16 techniques, therapies and techniques were identified: (1) visual biofeedback, (2) virtual reality, (3) upper limb interventions (constraint-induced movement therapy, hand–arm intensive bimanual training and augmenting occupational therapy), (4) trunk targeted-training interventions (with vibrating platform), (5) treadmill training (both with and without partial body weight support), (6) reactive balance training, (7) progressive resistance exercises, (8) the Perfetti method, (9) the Doman–Delacato method, (10) the Padovan method, (11), reflex locomotion: Vojta therapy, (12) neurodevelopment therapy: Bobath approach, (13) hippotherapy stimulators (horse-riding based on robotics), (14) hippotherapy (real horse-riding), (15) gross motor task Training, and (16) functional electrical stimulation.

The survey, based on the previously identified sixteen treatments, firstly showed the demographic characteristics of the participants as seen in Table 1.

The results on the knowledge of each one of the listed treatments are shown in Figure 1. The best-known treatments (more than 90% of the participants knew them) were: treadmill training, the Perfetti method, reflex locomotion: Vojta therapy, neurodevelopment therapy: Bobath approach, and hippotherapy. In contrast, the lesser-known treatments (more than 20% of the participants did not know them) were: visual biofeedback, reactive balance training, the Doman–Delacato method, the Padovan method and hippotherapy stimulators. By far, the most unknown treatment was the Padovan method (60.5%).

The results on the degree of use of each one of the listed treatments are shown in Figure 2. The treatment with the highest degree of use was neurodevelopment therapy: Bobath approach (39.5% and 27.9% of the participants always or very often, respectively, used it). Other very commonly used (always, very often) treatments were: upper limb interventions (11.6%, 41.9%), reactive balance training (23.3%, 32.6%), progressive resistance exercises (14.0%, 30.2%) and gross motor task training (27.9%, 41.9%). Treatments with a higher percentage of a medium degree of use (sometimes) were: the Perfetti method (39.5%) and virtual reality (27.9%).

On the other hand, a relatively high percentage of treatments were never used (more than 50% of the participants never had used them): trunk targeted-training interventions, the Doman-Delacato method, the Padovan method, hippotherapy stimulators and hippotherapy. Moreover, there were a couple of treatments that were rarely used (never, almost never): reflex locomotion: Vojta therapy (41.9%, 14.0%) and functional electrical stimulation (48.8%, 32.6%).

It is important to consider that the options ‘Never’ and ‘N/A’ regarding the degree of use of each one of the listed techniques, therapies, and interventions, were not only selected by the participants who did not know them, but also by some of the participants who knew them. That is why the lower limit for the percentage of participants who knew each one of the listed treatments but did not use them was obtained.

The results shown in Figure 3 indicated that more than 60% of the participants who knew of hippotherapy and the Doman–Delacato method never used them. The fact that that more than 25% of the participants who knew of trunk targeted-training interventions, reflex locomotion: Vojta therapy, hippotherapy stimulators and functional electrical stimulation never used them either is remarkable. On the contrary, all the participants who knew of reactive balance training, neurodevelopmental therapy: Bobath approach and gross motor task training used them with different frequency.

The results on the level of satisfaction of each one of the listed treatments are shown in Figure 4. On the one hand, the treatments which showed the highest levels of satisfaction (very satisfied, satisfied) were neurodevelopmental therapy: Bobath approach (39.5%, 34.9%), reactive balance training (25.6, 46.5%) and gross motor task training (20.9%, 60.5%). There were other treatments with good levels of satisfaction, such as upper limb interventions (18.6%, 44.2%), progressive resistance exercises (18.6%, 48.8%) and the Perfetti method (9.3%, 39.5%). On the other hand, some of the treatments which showed high percentages of medium levels of satisfaction (neither satisfied nor dissatisfied) were reflex locomotion: Vojta therapy (20.9%) and functional electrical stimulation (16.3%). Finally, the treatment which showed lower levels of satisfaction (very dissatisfied, dissatisfied) was the Doman–Delacato method (4.7%, 14.0%).

According to the previous data, the bivariate correlation (Pearson correlation coefficient, ρ) and the corresponding scatter plot and regression line were obtained to compare regularly used treatments (sum of ‘always’ and ‘very often’ percentages for each one of the listed treatments) and treatments with a high level of satisfaction (sum of ‘very satisfied’ and ‘satisfied’ percentages for each one of the listed treatments). According to these results, there was a strong positive correlation (ρ = 0.92) between the most used treatments and elevated levels of satisfaction, which could be corroborated by the corresponding regression line (see Figure 5).

The results concerning the opinion of the participants (expressed in an open-ended question) about the treatments with a more promising future in the rehabilitation of patients with CP show, on the one hand, that many of them opted for the Bobath approach in combination with some other techniques such as reflex locomotion: Vojta therapy, the Perfetti method, virtual reality or biofeedback. On the other hand, a few participants named other techniques which were not considered in this study, such as Therasuit, myofascial structural integration, kinesio-taping, and proprioceptive neuromuscular facilitations.

In addition, a series of points from the final observations and comments of the participants should be highlighted: the need for a multidisciplinary approach in the rehabilitation processes of children with CP, the lack of facilities and equipment in many health centers to carry out the more innovative rehabilitation techniques, the need for new methods based on sports, games or dance to improve postural control, and the proposal of using classic techniques with new and technologically advanced approaches.

## 4. Discussion

This discussion analyses the previous results and aims to determine if there is a correspondence between clinical practice preferences and the scientific evidence supporting the effectiveness of each treatment.

To begin with, it is important to highlight that some of the best-known treatments (treadmill training, reflex locomotion: Vojta therapy, neurodevelopment therapy: Bobath approach, and hippotherapy) are also the most well-established methods in clinical practice, not only in the field of CP rehabilitation but also in other rehabilitation areas (i.e.,: Parkinson’s disease, sclerosis, stroke and hemiplegia). In fact, there is an abundance of publications in recent years describing them, as well as their applications, pros and cons, effects, improvements, evolution, and associated results [15,24,25,26,27,28,29,30,31,32,33,34,35]. A deeper insight into these methods will be carried on later in this text.

On the other hand, the lesser-known methods (visual biofeedback, reactive balance training, the Doman–Delacato method, the Padovan method and hippotherapy stimulators) deserve individual analysis.

Regarding biofeedback (not only visual but also audio and/or haptic biofeedback), it is important to highlight that even though there are several studies that have researched biofeedback-based interventions for people with CP and other neuromotor diseases, there is still no consensus regarding the correct implementation of biofeedback mechanisms [36]. This lack of unanimity could explain the frequent non-implementation of this method in clinical practice. Therefore, it is important to consolidate current evidence to direct future study to develop effective biofeedback rehabilitation strategies. With that said, the use of visual biofeedback within virtual reality environments will be discussed later on this text.

The reactive balance training methods were traditionally used by fitness instructors, but since these kinds of exercises may improve the control of reactions and have the potential to prevent falls after losing balance in daily life, the technique is being tested in children with CP. According to our results, those clinicians who use these methods in children with CP are mostly satisfied with them. Regarding the evidence for its effectiveness, while one recent level II evidence study concluded that although postural control is modifiable, and does improve in response to intense balance training, more research is necessary to determine the type and frequency of intervention needed to impact postural control in these patients [37]. Other studies (level II) claimed that training using the Biodex New Balance System^TM^ SD (three times per week for 12 weeks) improved the limits of stability and standing balance and reduced fall risk in children with diplegic CP [38,39]. From a clinical point of view, balance is one of the main impairments for children with spastic diplegia [40], and the results using this kind of therapeutic tool showed a possible intervention with a robust scientific basis. Therefore, deeper research should be conducted following the use of this system.

The Doman–Delacato method (also known as patterning) is a classic procedure claiming that passively repeating steps in normal development can overcome brain injuries. However, the absence of a neuroscientific basis, the inconsistent results of uncontrolled studies over the years, and the lack of controlled trials resulted in the method not being recommended for many years [28,41,42,43,44] This can explain why the method is not as well-known as others, or why it is not used among those who know it.

The Padovan method, also known as the neurofunctional reorganization method, and supported on the plasticity of the nervous system, is based on the passive repetition of a sequence of neuroevolutionary movements until the patient has the autonomy to perform them actively. Although there are a few studies claiming the effectiveness of the method in premature and term infants who suffered perinatal hypoxia [45,46] there is a lack of high-level evidence studies presenting results in children with CP or related neuro-motor problems. This fact, as well as its holistic approach, usually considered as a pseudo-therapy, can explain why the method is, by far, the less-known method among the clinicians.

Hippotherapy is a well-known and well-established technique, but its degree of use is not as high as expected. One possible reason is that, despite its benefits, it is restricted by availability, high cost and safety factors [47]. These restrictions are supposed to be the main reasons why hippotherapy stimulators, robotic devices that use the oscillatory action of the seat to stimulate the horse-riding experience, are recently being proposed as an alternative method for the classic hippotherapy method. Although in recent years, according to some pilot/case studies [48,49], the use of these devices has been reported to have beneficial effects on spasticity, postural control, and motor function in children with CP, there is still a lack of high-level evidence supporting their efficiency [8]. Both the novelty of these platforms and the need for further research to establish their real potential, and consequently, the fact that their use is not widespread, would be the reasons why they are not yet well known by some of the clinicians working with CP patients.

Regarding the degree of use of the different techniques, therapies, and interventions, it is remarkable that there are several methods which are not commonly used among those who know them. The possible reasons behind this for some of the methods (Doman–Delacato, hippotherapy, and hippotherapy stimulators) have already been discussed, but others (trunk targeted-training interventions, reflex locomotion: Votja therapy or functional electrical stimulation) need a deeper insight.

Trunk-targeted training involves exercises aimed at improving trunk muscle strength and control and, in recent years, some studies suggest that although the technique should be further investigated, it could be a potentially effective treatment for children with poor trunk control [8,50]. According to this, the novelty of the application of this technique would justify its lack of use in the clinical practice.

Votja therapy is a dynamic neuromuscular treatment method based on the developmental kinesiology and principles of reflex locomotion but, although quite a few recent case studies [51], controlled and experimental studies [29,52,53], are available and present some moderate positive results, there is a lack of high-level evidence for justifying the usage and effectiveness of Vojta therapy [15], which could explain why, according to the presented results, a number of the clinicians who know the treatment do not use it.

Functional electrical stimulation (FES), also known as neuromuscular electrical stimulation (NES), is a therapy utilizing advanced computer technology to deliver electrical stimulation to paralyzed or weak muscles. FES was originally developed to help people with paralysis but is now being used to treat quite a few medical conditions. However, according to recent scientific publications, there is no high level of evidence of the effectiveness of this therapy on its own [54], but there are some promising results from high-level evidence publications suggesting that FES might be used as an adjuvant therapy to improve gross motor function in children with spastic CP [55,56,57]. Nevertheless, further research is still necessary.

On the other hand, as presented previously, the techniques, therapies and interventions that were used always or used very often with a higher percentage were neurodevelopment therapy: Bobath approach, upper limb interventions, reactive balance training, progressive resistance exercises and gross motor training.

Among these, neurodevelopment therapy: Bobath approach was, by far, the therapy which was said to be always used by a higher percentage of the respondents. The Bobath concept, which emphasizes both the integration of postural control and task performance, and the selective movement control to produce coordinated sequences of movements, is, as mentioned before, one of the most well-established problem-solving approaches in the field of rehabilitation of neuro-motor dysfunctions [58,59,60]. In addition, it is important to highlight that, although there are some authors that highlight the effectiveness of this therapy claiming that it improves functional motor ability, independence level on daily living activities, and balance ability in children with CP [31,61], some recent systematic reviews concluded that, contrary to popular beliefs (in fact, this treatment obtained the highest percentages of use in the survey), the Bobath approach is not backed sufficiently by evidence [62] and that there are no circumstances where any of the aims of neurodevelopment therapy could not be achieved by a more effective treatment [15]. These results demonstrated a controversy that should be tackled with further research.

Upper limb interventions (constraint-induced movement therapy, hand–arm intensive bimanual training and augmenting occupational therapy), which are focused on helping children with hemiplegic CP mainly learn to use both hands together to complete everyday activities, also showed relatively high percentages regarding frequency of use. The fact that of those children with cerebral palsy, 80–90% present hemiplegic cerebral palsy [63], and the fact that some randomized controlled trials [64,65] and systematic reviews/meta-analysis [15,66] suggested that these kinds of interventions are more effective than standard care, could explain this result.

Regarding the reactive balance methods, it is important to highlight that although some participants did not know them, those who knew and used them were mostly satisfied with the obtained results as discussed before, which can explain why these methods are among those with higher frequency (always/very often) of use.

With regard to progressive resistance exercises, there is a certain degree of controversy: although the method is commonly used in clinical practice [67], which agrees with the presented results, several studies claimed that even though these interventions could be effective in increasing muscle strength, improving balance and gait of even speed, more rigorous studies are needed to determine the real contribution to gross motor function [68,69,70]. Others, however, claim that strength training programs have positive functional and activity effects on gross motor function (without increasing spasticity) when adequate dosage and specific principles are used [71].

In addition, it is remarkable that the analysis for gross motor task training also shows a gap, because although the kind of interventions proposed by this method are commonly used in children with CP [13], there are some recent scientific publications claiming that there is limited evidence to support task-specific gross motor skills training. Moreover, recommendations for use over this method are limited by poor study methodology and heterogeneous interventions [72,73,74]. In summary, despite health professionals often prescribing gross motor activities to people with neuro-motor disabilities, primarily to improve function, there is no comprehensive evaluation of the evidence for the effectiveness of this kind of intervention in people with CP [13].

Next, after discussing the results regarding the degree of knowledge and frequency of use of the techniques, therapies and interventions identified for this study, it is important to analyze the results regarding the associated degree of satisfaction.

On the one hand, as previously shown in the results, there was a strong positive correlation (0.92) between the most used treatments, already discussed, and elevated levels of satisfaction as expected: it is reasonable that health professionals use in their daily clinical practice those solutions they are most satisfied with according to their previous experience and perception. This could explain why neurodevelopment therapy: Bobath approach, reactive balance training, or even progressive resistance exercises and gross motor task training, despite the controversy regarding their level of evidence commented on above, are the options that, along with upper limb interventions, are associated with a higher level of satisfaction.

On the other hand, there are other solutions whose levels of satisfaction are also remarkable but, on the contrary, are not as commonly used in clinical practice as the previous ones. The following methods are included within this group: the Perfetti method, virtual reality and visual biofeedback.

The Perfetti method, also known as cognitive sensory motor training, is a motor learning model that emphasizes high-level cognitive function thanks to the integration of perception–cognition–activity processes [75]. Even though there is no strong evidence of the effectiveness of the more traditional version of this method in patients with CP, [76] new cognitive sensory motor training systems based on videogame platforms or even virtual reality are currently being developed. It is important to highlight that, although the state of the technique is still in the early stages and further research is needed, the authors of some preliminary studies based on these new Perfetti-based platforms claimed that the use of customizable digital and virtual environments with both visual and auditory feedback produce comfortable training sessions that might improve the results in patients with different neuromotor disabilities [77,78,79]. This, along with the fact that these kinds of systems are not widely available yet, could explain our results.

In fact, in recent years, several solutions based on both biofeedback, that is the technique of providing biological information to patients in real-time, and on virtual reality were developed not only as a novelty associated with the Perfetti method or as a complementary option for other therapies, but as a real and complete alternative of rehabilitation in children with CP [80]. Clinicians can now design virtual environments to achieve a variety of therapeutic objectives by varying the task complexity, type, and amount of feedback [81]. Although the current evidence is still weak [82], recent preliminary results suggest that therapies based on virtual reality and biofeedback (visual, auditory and/or electromyography, among others) may improve motor function in the upper and lower extremities in children with CP [48,83,84]. In addition, it is important to highlight that demonstration of the effectiveness of this kind of intervention depends on the degree to which the attained skills transfer to the “real world” [82,85]. Again, the lack of availability of this type of system in healthcare facilities because of its novelty and, in this case, probably its cost, along with the current lack of evidence, may explain the presented results. Nevertheless, the use of these systems seems to be spreading, since, as commented before, the results are promising and agree with the level of satisfaction of the participants of this study.

On the other hand, only a moderate uphill relationship was found between the percentage of participants who knew but never used each one of the studied treatments and elevated levels of dissatisfaction. In this case, two extreme cases deserve a detailed analysis: hippotherapy and the Doman-Delacato method.

Although both solutions were known but not used in more than 60% of the cases, the first one showed no dissatisfaction results while the second one showed the highest level of dissatisfaction (18.7%) registered among the participants. Regarding hippotherapy, despite those who tested the technique being satisfied with the results, its restricted availability, high cost and other logistical and safety factors can explain its lack of use as commented before [47]. In contrast, as shown previously, the Doman-Delacato method is not usually recommended as it did not show the expected benefits for the patients despite being a classic and relatively well-known procedure [36,41,42,43,44]. This can explain the remarkable degree of dissatisfaction registered by the survey.

After the previous analysis, there is still one technique among those considered for the study which has not been analyzed yet: treadmill training. A typical form of gait training has been performed traditionally with assistive devices or parallel bars. The treadmill has recently gained more attention as an instrument for gait training and assessment, and it is used for children with CP to help them to improve balance and build the strength of their lower limbs [80]. In fact, there are moderate-to-positive results supporting the effectiveness of this kind of training in children with CP [82,86,87]. Moreover, there are also promising results regarding the effectiveness for improving gait, balance, muscular strength, and motor gross function in children with CP with rehabilitation programs based on treadmill training and virtual reality [88]. In addition, it is important to highlight that a very recent meta-analysis concluded that treadmill training on CP was effective for gait endurance, gait speed and limb support time than cadence and step length [89].

To round up this discussion, we also considered other techniques, such as Therasuit, myofascial structural integration, kinesio-taping, and proprioceptive neuromuscular facilitations that, while not considered for the study, were brought up by the participants.

The Therasuit method is based on an intensive and specific exercise program aimed at eliminating pathological reflexes and establishing new and functional patterns of movements. This solution uses, among other tools, a soft dynamic proprioceptive orthosis (Therasuit^®^) that aligns the body as close to normal as possible. The creators of the method described it as the “new standard treatment for cerebral palsy” [90] and highlighted the effectiveness of this solution regarding the levels of functional improvements and the progress in coordination, strength, balance, range motion and movement control [91]. However, other studies (levels I and II) suggested caution in recommending the use of these therapeutic suits as they found very low-quality evidence regarding the activity outcomes and the proposed protocols [90,91,92,93,94,95]. Therefore, in line with the discussion about progressive resistance exercises and gross motor task training, more research is necessary to clarify the true impact of the Therasuit method.

Some preliminary studies have recently claimed that myofascial structural integration (MSI) combined with conventional rehabilitation can be effective in reducing spasticity and improving range of motion, strength, and function of the upper limb in children with CP, compared to conventional rehabilitation by itself [96,97,98]. However, the lack of evidence-based literature shows that it is still early to consider this option as a treatment tool for children with CP [15]. Still, it should be noted that the underlying mechanism of manual therapy techniques could be useful for pain management [99].

Kinesio-taping is a relatively new therapeutic tool used in rehabilitation programs of children with CP, although it has been in use for a long time in the sport and orthopedic fields. Kinesio tape is a specialized elastic-like tape designed to mimic the elastic properties of the muscle, skin, and fascia, leading to enhancing proprioception, diminishing pain and edema, reducing muscle spasms, and strengthening the muscles [100,101]. Some studies aimed at determining the impact of this tool conclude that kinesio-taping must be coupled with other rehabilitation techniques [102]. For example, gross motor function and excessive rounding of the spine (kyphosis) in children with CP can, according to a recent study [102], be significantly improved with kinesio-taping and neuromuscular electrical stimulation, in addition to the conventional rehabilitation. However, there is a lack of highly methodological studies about the efficacy of the method in children with CP, so randomized control trials with well-established protocols are needed to increase the confidence in the application of kinesio-taping in this context [103,104,105].

Proprioceptive neuromuscular facilitation (PNF) training (also known as the Kabat concept) is a motor learning approach that targets the improvement of proprioceptive function, focusing on the use of somatosensory signals to improve or restore the sensorimotor function. According to recent research, there is converging evidence that PNF training, not only based on its classical approach but also on more innovative approaches, could yield meaningful improvements in somatosensory and sensorimotor function. Such approaches include the use of virtual reality systems as proprioception methods [106] or simultaneous proprioceptive and visual biofeedback platforms as sensory integration training environments [107]. However, there is still no strong evidence of its effectiveness so there is a clear need for further work [108].

To summarize this discussion, it is important to highlight its main ideas as well as describe its possible implications in education, clinical practice, and research, derived from the previous analysis:

On the one hand, it is remarkable that some well-known methods, such as Votja therapy, the Doman–Delacato method, or the Bobath approach, that have been used for years in clinical practice, do not have a solid level of scientific evidence to support their use. According to our preliminary results, the level of satisfaction for the first two methods is relatively low (especially for the Doman–Delacato method), but, surprisingly, the levels of satisfaction registered for the Bobath approach are the highest among all the proposed techniques. These results suggest that the education of clinical professionals in this context could lack the scientific rigor that should be expected, transmitting knowledge about techniques that have traditionally been used without prioritizing those that are effective, based on scientific evidence, or those emerging techniques whose level of evidence needs to be demonstrated with further research. This fact calls into question the rigor of some interventions carried out in clinical practice and opens the door to the need to introduce new protocols in the training of professionals with scientific evidence and objectivity as the fundamental pillars. In fact, awakening the scientific interest of students, as well as the motivation to stay updated, could lead to breaking the current gap, which, on the other hand, would help to classify and determine the level of evidence of those emerging techniques, mainly supported by the use of new technologies (such as those based on virtual reality or acoustic/visual biofeedback), that, according to preliminary studies, show promising results in the rehabilitation of children with CP.

In short, the application of any method in clinical practice should always be supported by solid scientific evidence. The scientific community already offers relevant studies such as [8,15,22], among others, that could be used as new guidelines in education, clinical practice, and research.

## 5. Conclusions

The existing gap between the scientific evidence and the clinical practice in physical therapies for cerebral palsy presented in this paper is a first step in the objective of constructing an evidence-based physical therapy practice for children with CP. Despite the limitations of having a small sample, this work contributes to identifying the importance of being aware of the need to use scientific methodology to measure the effectiveness of the treatments that centers use in their daily routines. By following the therapeutic approaches supported by the literature, the clinical practice could become less reliant on the individual therapist’s skill and more robust. All in all, it is important to highlight that there is a pressing need for functionally based research in the clinical practice that uses objective protocols and outcome measurements to evaluate the interventions on children with neuromotor disorders.

## Figures and Tables

**Figure 1 ijerph-19-14535-f001:**
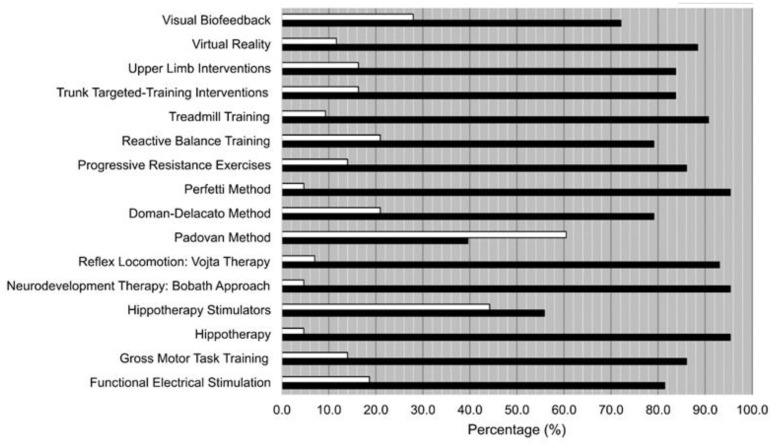
Knowledge of each one of the techniques, therapies and interventions listed on the survey. No (white bars)/Yes (black bars).

**Figure 2 ijerph-19-14535-f002:**
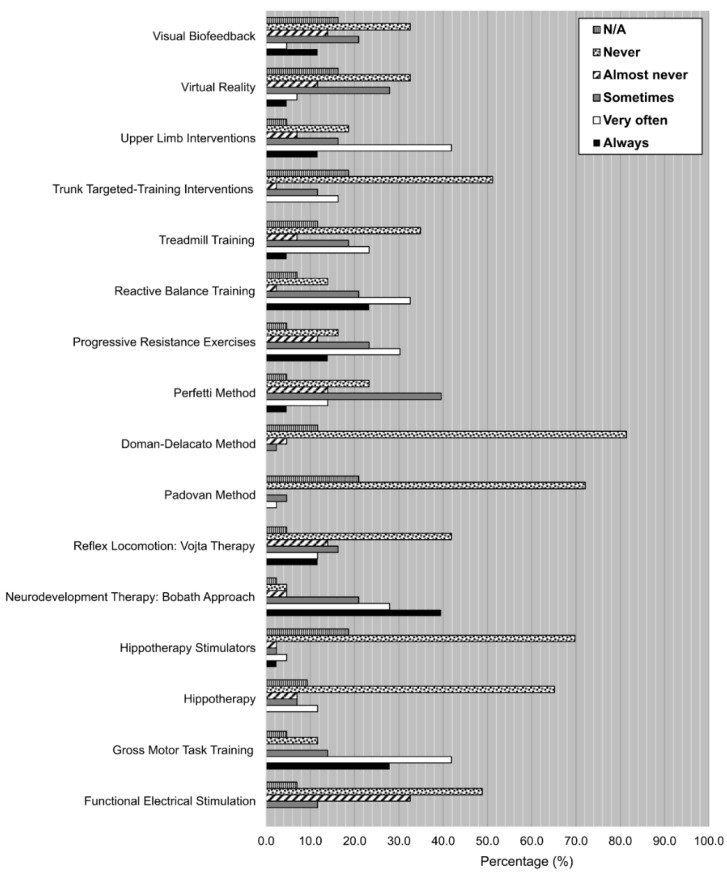
Degree of use (%) for each one of the techniques, therapies and interventions listed on the survey. Scale: Always, very often, sometimes, almost never, never, N/A.

**Figure 3 ijerph-19-14535-f003:**
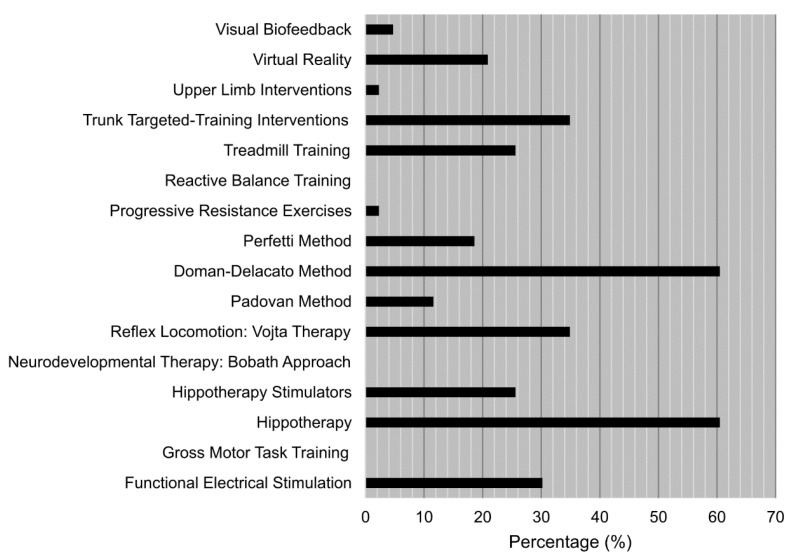
Lower limit (%) of participants who knew each one of the listed treatments but never used them.

**Figure 4 ijerph-19-14535-f004:**
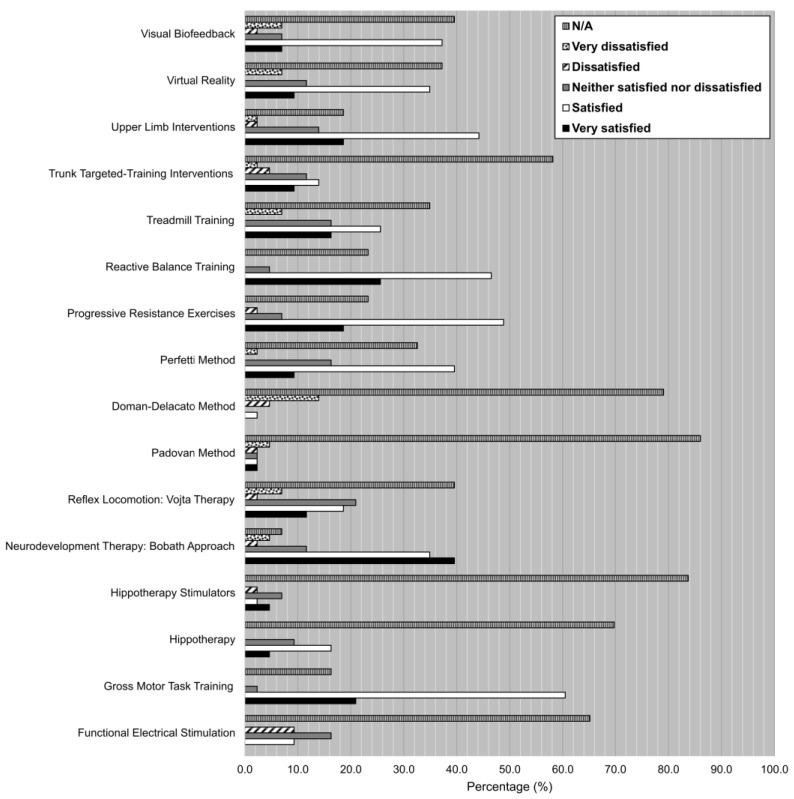
Degree of satisfaction (%) for each one of the techniques, therapies and interventions listed on the survey. Scale: very satisfied, satisfied, neither satisfied nor dissatisfied, dissatisfied, very dissatisfied, N/A.

**Figure 5 ijerph-19-14535-f005:**
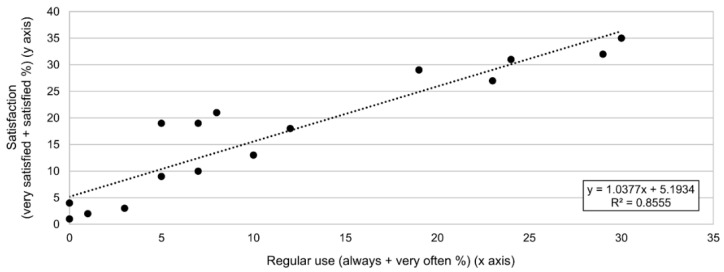
Scatter plot and regression line for ‘Regular use’ vs. ‘Satisfaction’ variables.

**Table 1 ijerph-19-14535-t001:** Demographic characteristics of the participants, including current occupation, age, and years of experience in the field (*n* = total number of participants).

Characteristic	Number (%) of Participants *n* = 43
**Current occupation**	
Physiotherapist/Pediatric physiotherapist	35 (81.4)
Occupational/Neuro-psychomotricity therapist	8 (18.6)
**Age**	
<25	2 (4.7)
25–29	14 (32.6)
30–34	9 (20.9)
35–39	10 (2.3)
≥40	8 (18.6)
**Years of experience in the field**	
<5	10 (23.2)
5–9	12 (27.9)
10–14	7 (16.3)
15–19	7 (16.3)
≥20	7 (16.3)

## Data Availability

The data presented in this study are available on request from the corresponding author.

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
