# Peer review of "Studying the Research–Practice Gap in Physical Therapies for Cerebral Palsy: Preliminary Outcomes Based on a Survey of Spanish Clinicians"

_ijerph, 2022, doi:10.3390/ijerph192114535_

Round 1
Reviewer 1 Report (Previous Reviewer 2)
It is necessary to present the concept of ICF and the reason for presenting the GMFCS in the introduction in detail. What is the relationship to this study?
The reason for describing the characteristics of children with cerebral palsy in the introduction also needs to be supplemented. The short circuit and connection of the research-actual gap described later seems to be lacking, and it seems that the flow does not fit logically.
A detailed explanation of how the subjects were selected for the study should be added. What process did 43 people go through?
The method of composition of the survey site should also be described. Did you refer to existing literature? How was the validity of the items checked?
Discussions are listed by individual intervention. It seems necessary to come up with a plan that can be described by dividing it into areas or types rather than by individual interventions. In particular, I think it is necessary to summarize and present the implications to be reflected in clinical, education, and research.
Author Response
Please see the attachment.

Reviewer 2 Report (New Reviewer)
Thank you for the invitation to review this article, but it raises some doubts before publication.
Especially the fact that it is a survey of physiotherapy techniques that have been evaluated by other profiles; together with the low sample of participants, so it is not possible to extrapolate the current reality of this country
Therefore, it should be specified why the response from occupational therapists was accepted.
Very interested that you can add the questions in an appendix
Other comments
Abstract
· No use of acronyms.
Introduction
· Talk about unilateral and bilateral, according to the current change.
· Delete sefip’s link
Methods
· Because those years of searching, based on what?
· You name OT but you don't search in OTSeeker
· Because only in English, it is a bias
· From lines 133 to 144……….it would be necessary to synthesize
· Age data would not go here (results)
· Sample is small…..be careful with what is said
Results
· Difference between hiptotherapy?
· It would be interesting to be able to break down upper limbs interventions, since HABIT is better than CIMT (Hoare’s Cochrane SR and Meta-Analysis)
Discussion
· Padovan is considered a pseudotherapy, so you have to be very careful with what you say.
· Novak's Kinesiotaping references should be reviewed, since they say something different from what is expressed in his manuscript.
· Regarding strength, there are two meta-analyses that could be of help. I write references
· Park EY and Kim WH. Meta-analysis of the effect of strengthening interventions in individuals with cerebral palsy. Res Dev Disabil 2014; 35(2): 239–249.
· Merino-Andrés J, García de Mateos-López A, Damiano DL, Sánchez-Sierra A. Effect of muscle strength training in children and adolescents with spastic cerebral palsy: A systematic review and meta-analysis. Clin Rehabil. 2022;36(1):4-14.
Conclusions
· It cannot be extrapolated with such a small sample, text would have to be modified to fit that
Author Response
Please see the attachment.

Reviewer 3 Report (New Reviewer)
Where are the studies of Stergiou?
Survey- identify kind of validation was made and how
References- with a lot of small incorrections (absence of authors; absence of bold in year; repeated commas,...)
Round 2
Reviewer 1 Report (Previous Reviewer 2)
Even if we accept the claim that the authors have succeeded in classifying the PI, I think that the restructuring of the discussion presented by the reviewer can be sufficiently reflected. It is regrettable that this is not reflected. Considering that it is a preliminary study does not mean that it is not necessary to present specific implications for the study results.
Author Response
Please see the attachment.

Reviewer 2 Report (New Reviewer)
Thank you for all your changes and for your comments. I think that manuscript is better than the first one. But I want write you some suggestions and comments.
- If your research is a preliminary study, you could think if you write it in the title
- I understand your comments about include OT in your survey, so I think that it's important that you write it in methods. For example, "we include OT because they work in team with PT (or similar sentence)
- I think that it's important that you include the survey as appendix
- I think that it's important that you write that you selected these years to search studies in base Novak's research
- About KT's references, there are some more about systematic reviews in Novak's research (2020)
A comment: spanish law allow work the function only PT, and allow work the activity and the participation only OT.
Regards
Author Response
Please see the attachment.

Reviewer 3 Report (New Reviewer)
No further comments. Congratulations.
Author Response
Please see the attachment.

This manuscript is a resubmission of an earlier submission. The following is a list of the peer review reports and author responses from that submission.
Round 1
Reviewer 1 Report
Well-presented manuscript, attempting to analyze the existing gap between research and practice gap in physical therapies centered on Cerebral Palsy. I would like to mark some points, relevant to your survey:
You have mentioned that ‘The level of evidence was not an exclusion criterion for this first step. This decision was taken because we did not want to dismiss methods having low levels of evidence. On the other hand, it is important to highlight that the discussion on the differences between the clinical practice preferences and the scientific evidence of effectiveness regarding those previously identified methods, is mainly supported by studies of level I or II evidence. Levels III to IV were included only if highest level of evidence did not exist on the topic.’ It seems that there is inconsistency between these two methods of selection of evidences, regarding the inclusion criterion and the discussion of the differences, respectively.
The number of participants is relatively low. Apart from that, there is no data regarding the rehabilitation center they were working for, the number of patients they have treated and the clinical severity of the patients to whom they offered their services.
Finally, and most importantly, this survey is limited to Spanish Clinicians. Although it is relatively well-conducted, it would be of enhancing value to compare your data from similar outcomes coming from other countries. This would offer data that are not restricted to the boundaries of any specific country and would be more worldwide acceptable and interesting to interpret.
Reviewer 2 Report
Dear Authors
Thank you for the opportunity to review this paper.
The description of cerebral palsy fills much of the introduction. It is necessary to abbreviate this part and expand the description of the necessity and rationale of this study.
Specific research question should be stated.
More detail information was needed. For example, sex, their specific field.
Review process should be described more detail. How many studies was searched at initial search? Was there any double check for selecting the studies or categories? How many studies was categorized in each technique?
It is necessary to add statistical analysis results according to the characteristics of the respondent. For example, among the interventions listed in the survey, there seems to be one that is frequently used in physical therapy and occupational therapy, respectively. Therefore, it seems necessary to examine the interventions that differ between each specialized area and those that are not, and if there is a difference, it is necessary to present the results of whether the characteristics of the intervention are different for each treatment area or not. It is thought that describing the discussion based on these results can suggest more specific implications.
Also, depending on the age, knowledge of interventions may differ, and this needs to be explored.
Typo error
- Cerebral Palsy --> cerebral palsy
- PT (Physical Therapy) --> Physical Therapy (PT)
Round 2
Reviewer 1 Report
I have carefully studied the cover letter that was accompanying the revised version of your manuscript. I would like to mention that no significant corrections were added in an effort to make a counter-argument to the remarks that were submitted towards your side.
Reviewer 2 Report
Dear authors
It is regrettable that most corrections to the comments have not been reflected.
Your response: As the purpose of the study is not focused on comparing the responses of the survey according to the characteristics of the respondents, we did not add this kind of information beyond the basic demographic information presented in the text. In addition, we did not add the names of the institutions and rehabilitation centers where the respondents work to ensure the anonymity of both the centers and the participants in the survey. The reason is, as stated in the text, that a certain level of controversy has emerged from the analysis of the results.
--> Although survey did not focused on comparing the responses of the survey according to the characteristics of the respondents, basic information about characteristics of the respondents should be provided to generalization of the results of this study. Generalization is major purpose of the study, as you know.
Your response: We added more detailed information on the review process within the text (lines 114 to 122). We have also added the figures that show the number of studies included in our initial search (line 179). As many of the identified studies were systematic reviews, with several interventions depicted in each of them, a subclassification for each kind of intervention was not carried out. A double check for selecting the studies was not made.
However, we want to emphasize that, although the identification of the interventions and the subsequent discussion required a review process, the article itself is not a standard systematic review, and this is the reason why the review process did not conforms to the standard model (for example, PRISMA) It has been adapted to the specific needs of this study instead; mixing the review with the analysis of the survey allows for the comparison of the literature with the reality of clinical practice.
--> Although your study mixed the review with the analysis of the survey allows for the comparison of the literature with the reality of clinical practice adapted to the specific needs of this study instead, review should be performed systematically. Double check should be performed for validity and reliability of this study.
Your response: Most of the respondents work in multidisciplinary centers, where the strategies used for the performance of the daily clinical practice are the result of collaborative work among different healthcare professionals. We believe that, within this context, there is no reason for making a comparison based on the characteristics of the participants.
--> Although most of the respondents work in multidisciplinary centers, they have major role in the centers. The results might be different according to their major role.